

# Genotypic characterization and genome comparison reveal insights into potential vaccine coverage and genealogy of *Neisseria meningitidis* in military camps in Vietnam

Trang Thu Le[1], Thach Xuan Tran[1], Long Phi Trieu[2], Christopher M. Austin[3,4], Huong Minh Nguyen[1] and Dong Van Quyen[1]

[1] Laboratory of Molecular Microbiology, Institute of Biotechnology, Vietnam Academy of Science and Technology, Hanoi, Vietnam
[2] Laboratory of Microbiology, Military Institute of Preventive Medicine, Hanoi, Vietnam
[3] Deakin Genomics Centre, Deakin University, Geelong, Victoria, Australia
[4] Centre for Integrative Ecology, School of Life and Environmental Sciences, Deakin University, Geelong, Victoria, Australia

Corresponding authors
Huong Minh Nguyen,
huongminh.nguyen@ibt.ac.vn,
nguyen.huong.m@gmail.com
Dong Van Quyen, dvquyen@ibt.ac.vn,
dvquyen@gmail.com

## ABSTRACT

**Background**. *Neisseria meningitidis* remains the main cause of sporadic meningitis and sepsis in military camps in Vietnam. Yet, very limited molecular data of their genotypic and epidemiological characteristics are available from Vietnam, and particularly the military environment. Whole genome sequencing (WGS) has proven useful for meningococcal disease surveillance and guiding preventative vaccination programs. Previously, we characterized key genetic and epidemiological features of an invasive *N. meningitidis* B isolate from a military unit in Vietnam. Here, we extend these findings by sequencing two additional invasive *N. meningitidis* B isolated from cerebrospinal fluid (CSF) of two meningitis cases at another military unit and compared their genomic sequences and features. We also report the sequence types and antigenic profiles of 25 historical and more recently emerged *N. meningitidis* isolates from these units and other units in proximity.

**Methods**. Strains were sequenced using the Illumina HiSeq platform, de novo assembled and annotated. Genomes were compared within and between military units, as well as against the global *N. meningitidis* collection and other isolates from the Southeast Asia region using PubMLST. Variations at the nucleotide level were determined, and phylogenetic relationships were estimated. Antigenic genotypes and vaccine coverage were analyzed using gMATS and PubMLST. Susceptibility of isolates against commonly used antibiotic agents was examined using E-test.

**Results**. Genome comparison revealed a high level of similarity among isolates both within and between units. All isolates showed resistance to chloramphenicol and carried identical *catP* gene with other Southeast Asian isolates, suggesting a common lineage. Their antigenic genotypes predicted no coverage by either Bexsero®or Trumenba®, and nucleotide variation analysis revealed diverse new, unassigned alleles at multiple virulence loci of all strains. Groups of singleton and unique novel sequence types extending beyond individual camps were found from epidemiological data of 25 other isolates. Our results add to the sparse published molecular data of *N. meningitidis*

in the military units in Vietnam, highlight their diversity, distinct genetic features and antibiotic resistance pattern, and emphasize the need for further studies on the molecular characteristics of *N. meningitidis* in Vietnam.

# INTRODUCTION

*Neisseria meningitidis* is an encapsulated Gram-negative bacterium that asymptomatically colonizes the human nasopharynx but can cause serious septicemia and meningitis upon entering the bloodstream and passing through the blood-meningeal barrier (*Rosenstein et al., 2001*). Carriage rate is age and setting dependent, among other factors, with high prevalence found in the age groups of infants (4.5%) and young adults (23.7%) (*Christensen et al., 2010*). Congregated living environment is another risk factor, as shown in increased carriage rate among university students living in dormitories (*Peterson et al., 2018*; *Breakwell et al., 2018*) or military recruits in camp sites (*Sim et al., 2013*; *Keiser, Hamilton & Broderick, 2011*). In these environments, meningococcal meningitis can sometimes become an outbreak, and sporadic cases often show recurring and cluster characteristics (*Peterson et al., 2018*). Hence, in many countries, preventative vaccination program is recommended for these high-risk groups (*Yezli, Wilder-Smith & Saeed, 2016*) and implementations have shown significant impact (*Broderick, Phillips & Faix, 2015*). However, to enable successful preventative implementation and reduce the risk related to these environments, as well as to aid future cases' diagnosis and treatment, it is crucial to have reliable and accurate monitoring data of sporadic cases and carriages in these groups.

Recent advances in genome sequencing technology allow for a significant volume of genomic data to be generated and made public. Such data also provide an unprecedented power of discrimination that is invaluable for studies of the relationship of closely related strains. The widely utilized *Neisseria* database PubMLST (https://pubmlst.org/neisseria/), at the time of preparation of this manuscript, hosts more than 22,000 genomes and 60,000 isolate records for *Neisseria*. Employing sequence data deposited to PubMLST, a core genome of 1605 loci was determined for *N. meningitidis* (*Bratcher et al., 2014*). Analysis of sequence variations at these loci has furthered our understanding of genomic variation within *N. meningitidis* population (*Harrison et al., 2017*) and enabled the study of closely related but distinct strains present in outbreaks (*Jolley et al., 2012*). Despite the usefulness of whole genome sequencing (WGS) analysis for epidemiological surveillance, data from Vietnam, and Southeast Asian region in general, are extremely limited. Previously, we have described for the first time the genome of a chloramphenicol-resistant invasive *N. meningitidis* B isolated from a military unit in Vietnam (*Tran et al., 2019*). Later, a study conducted by the Mahidol-Oxford Tropical Health Network (MORU) in Thailand, Laos and Cambodia identified eight additional *N. meningitidis* isolates carrying the identical chloramphenicol-resistant gene along other acquired resistance to multiple antibiotics

(*Batty et al., 2019*), suggesting the existence and expansion of a lineage in this region. Here we extend the previous studies by describing the genomic characteristics of two additional invasive *N. meningitidis* B isolated from CSF of two meningitis cases at another Vietnamese military unit in proximity with the previously reported, and conduct a global analysis utilizing the PubMLST database and associated analytical tools. We show that, besides the chloramphenicol non-susceptible lineage, reservoirs of known sequence types that cannot be assigned to lineages and, novel emerging sequence types make up a significant part of both invasive and carriage strains found in the military camp environments in Vietnam.

## MATERIALS AND METHODS

### Bacterial isolation and typing

All invasive *N. meningitidis* strains were isolated from CSF and all carrier strains were isolated from mouthwash samples at the Laboratory of Microbiology, Military Institute of Preventive Medicine, Hanoi as described previously (*Tran et al., 2019*). Two isolates, NMB_VN2013 and NMB_VN2015, were from the CSF of two confirmed meningitis cases discovered in 2013 and 2015 at Military Unit 2, a camp in the geographical closeness to Military Unit 1, the camp where the previously reported DuyDNT isolate was identified. Both cases were treated at the military hospital and subsequently recovered. Serogroup identification and multi-locus sequence typing (MLST) were either done according to previously described standard methods (*World Health Organization, 2011*) or by manually extracting the corresponding sequences from WGS data.

DuyDNT isolate was renamed NMB_VN2014 here for convenience and consistency. Available laboratory records of the Laboratory of Microbiology, Military Institute of Preventive Medicine, Hanoi were reviewed and a suitable collection of 25 isolates were chosen based on reported year (before and after 2013–2015), location (Military Unit 1, 2, and two nearby units here named Unit 3 and Unit 4) and availability of molecular characterization data (Table 1). Serogroup data, MLST data, *fHbp* and *porA* allelic variants were obtained from laboratory records. Case-related metadata, including year, clinical status, and location were obtained where available.

### Antibiotic susceptibility testing

Susceptibility of isolates to seven antibiotics, namely ampicillin, ciprofloxacin, cefotaxime, ceftriaxone, rifampicin, meropenem and chloramphenicol, was examined using E-test strip (bioMerieux, France) following the manufacturer guideline and MIC values were determined. Susceptibility was interpreted according to CLSI 2018 breakpoints (*CLSI, 2018*).

### Genome sequencing and analysis

Genomic DNA was extracted using GeneJET Genomic DNA Purification Kit (Thermofisher Scientific) in accordance with the manufacturer's instruction. Samples' quality was checked before sequenced using the Illumina HiSeq 4000 system (Macrogen). Genome assembly and annotation were performed as previously described (*Tran et al., 2019*). Annotated amino acid sequences were used to identify genes involved in antibiotic resistance (*Tran et*

**Table 1** Epidemiological characterization of historical and emerging *N. meningitidis* isolates from military camps in Vietnam.

| Isolate | Year[a] | Description[a] | Military unit[a] | Sequence type (ST)[b] | Strain designation[b] |
|---|---|---|---|---|---|
| Khoa | 2008 | Meningitis | Unit 2 | 1576 | B: P1.19,15-39: ST-1576 |
| 37C | 2012 | Carrier | Unit 3[c] | 4821 | C: P1.20,2: ST-4821 (cc4821) |
| 40C | 2012 | Carrier | Unit 3[c] | 4821 | C: P1.20,2: ST-4821 (cc4821) |
| Bach | 2013 | Meningitis | Unit 1 | 13455 | B: P1.19,15: ST-13455 |
| NMB_VN2013 | 2013 | Meningitis | Unit 2 | 1576 | B: P1.7-2,13: F4-6: ST-1576 |
| 14072 | 2014 | Carrier | Unit 4[d] | 13065 | B: P1.18, Δ: F18: ST-13065 |
| 14075 | 2014 | Carrier | Unit 4[d] | 13065 | B: P1. Δ, Δ: ST-13065 |
| 14089 | 2014 | Carrier | Unit 4[d] | 13065 | B: P1. Δ, Δ: F18: ST-13065 |
| 14155 | 2014 | Carrier | Unit 1 | 4821 | C: P1.20,2: ST-4821 (cc4821) |
| 14156 | 2014 | Carrier | Unit 1 | 4821 | C: P1.20,2: ST-4821 (cc4821) |
| 14157 | 2014 | Carrier | Unit 1 | 4821 | C: P1.20,2: F22: ST-4821 (cc4821) |
| 14196 | 2014 | Carrier | Unit 2 | 1576 | B: P1.7-2,13-2: ST-1576 |
| NMB_VN2014 | 2014 | Meningitis | Unit 1 | 13074 | B: P1.22-25,14-32: F4-6: ST-13074 |
| 15020 | 2015 | Carrier | Unit 1 | 1576 | B: P1.7-2,13-1: ST-1576 |
| 1513 | 2015 | Carrier | Unit 1 | 13056 | B: P1. Δ, Δ: F18: ST-13065 |
| 1523 | 2015 | Carrier | Unit 1 | 1576 | B: P1.7-2,13-2: ST-1576 |
| 1530 | 2015 | Carrier | Unit 1 | 1576 | B: P1.22-25,14: ST-1576 |
| 1533 | 2015 | Carrier | Unit 1 | 1576 | B: P1.7-2,13: ST-1576 |
| 1535 | 2015 | Carrier | Unit 1 | 44 | B: P1.7-2,13-1: ST-44 (cc41/44) |
| NMB_VN2015 | 2015 | Meningitis | Unit 2 | 1576 | B: P1.22-25,14: F1-7: ST-1576 |
| 1237C | 2016 | Carrier | Unit 3[c] | 4821 | C: P1.20,2: ST-4821 (cc4821) |
| 16005 | 2016 | Carrier | Unit 2 | 1576 | B: P1.22-25,14: ST-1576 |
| 16016 | 2016 | Carrier | Unit 2 | 1576 | B: P1.22-25,14: ST-1576 |
| 16406 | 2016 | Carrier | Unit 1 | 4821 | B: P1.7-2,14: ST-4821 (cc4821) |
| 16408 | 2016 | Carrier | Unit 1 | 4821 | B: P1.7-2,14: ST-4821 (cc4821) |
| 16416 | 2016 | Carrier | Unit 1 | 4821 | B: P1.7-2,14: F80: ST-4821 (cc4821) |
| 17088 | 2017 | Carrier | Unit 1 | 13074 | B: P1.22-25, Δ: ST-13074 |
| 17090 | 2017 | Carrier | Unit 1 | 13074 | B: P1. Δ, Δ: ST-13074 |

**Notes.**

[a]Isolates' metadata (year, clinical description, and location) were obtained from laboratory records of the Laboratory of Microbiology, Military Institute of Preventive Medicine, Hanoi.

[b]Molecular data was extracted from genomic sequence (NMB_VN2013, NMB_VN2014, NMB_VN2015) or laboratory records (other isolates). Strain designation was based on the previously recommended nomenclature (Jolley, Brehony, and Maiden 2007) , comprising of serogroup, porA type (Px), fHbp type (Fx), and sequence type (STx) (clonal complex (ccx)).

[c]Unit 3 is geologically close to Unit 1.

[d]Unit 4 is geologically close to Unit 2.

*al., 2019*). Additionally, allelic profile of relevant antibiotic resistant genes from PubMLST was extracted from WGS data of each genome, and where applied, PSI-BLAST was used to find homologous sequence. Antigenic profile, antibody cross reactivity prediction and allelic variants of virulent factors including capsular genes, Maf-toxin island, and outer membrane vesicle (OMV) genes were analyzed with gMATS (*Muzzi et al., 2019*) and the PubMLST server using default parameters.

## Genome comparison and phylogenetic analysis

Assembled genomes were submitted to the PubMLST website and allelic variants were automatically assigned for each locus. Genomes were compared at the seven loci of MLST scheme, the 53 loci of ribosomal MLST scheme, and the 1605 loci of the core genome cgMLST scheme for *N. meningitidis*. Allelic variants were further processed in Excel and where necessary, manually removed from comparisons. Genomes were aligned using the progressive Mauve software (*Darling, Mau & Perna, 2010*), and the phylogenetic distance between strains was determined at the whole genome level. Neighbor-Net networks were constructed from various comparisons implemented on the PubMLST and visualized by SplitsTree (*Huson, 1998*).

Raw sequence data of 18 Southeast Asian *N. meningitidis* was obtained from the European Nucleotide Archive (project PRJEB30968) (*Batty et al., 2019*), assembled using Spades V3.11.1 (*Bankevich et al., 2012*), and assembled contigs were used for phylogenetic network analysis by SplitsTree and other sequence comparisons.

# RESULTS AND DISCUSSION

## Characterization of NMB_VN2013 and NMB_VN2015 isolates
### Genome

Both NMB_VN2013 and NMB_Vn2015 had a genome size of ~2.1 Mb and ~51.2% GC content, and contained 2390 and 2409 CDS each, respectively. Each genome had 3 rRNA and 53 tRNA coding sequences and contained ~400 repetitive sequences. Overall, their genome size and content matched closely with the typical genome of previously reported *Neisseria* representatives such as *N. meningitidis* MC58 (*Tettelin et al., 2000*), Z2491 (*Parkhill et al., 2000*), FAM18 (*Bentley et al., 2007*), and *N. gonorrhoeae* NCCP11945 (*Chung et al., 2008*). Assembly data and genomic sequences of both genomes were deposited to NCBI Genomes database under BioProject ID PRJNA523495.

### Serogroup and Sequence type

Both NMB_VN2013 and NMB_VN2015 were serogroup B, as inferred by the presence of the *csb* gene from their genome sequence and confirmed by Vitek®. Multi-locus sequence typing (MLST) profiles extracted from WGS data grouped NMB_VN2013 and NMB_VN2015 into the same ST-1576, a singleton ST that had no known clonal complex. ST-1576 is closely related to ST-13074, which was assigned to NMB_VN2014 before, with two STs differed at a single locus (*aroE*). Polymorphic site analysis revealed 46 nucleotide changes and no deletion/insertion between the two alleles, *aroE* 9 of ST-1576 and *aroE* 4 of ST-13074.

### Antibiotic susceptibility

Previously, NMB_VN2014 was shown to carry a tetracycline (*rpsJ*) and chloramphenicol (*catP*) resistant genes. Identical *rpsJ* and *catP* genes were found in the genomes of NMB_VN2013 and NMB_VN2015. The 624 bp *catP* gene found in all three Vietnamese isolates was the same gene previously reported in France (*Galimand et al., 1998*), and Southeast Asia (*Batty et al., 2019*). Antibiotic susceptibility test confirmed NMB_VN2013

and NMB_VN2015's resistance to chloramphenicol, with the recorded MIC were 62 and 64 μg/ml, respectively (Table 2).

From WGS data, of the 11 antibiotic susceptibility genes analyzed by PubMLST, eight were identical in all isolates, including *gyrA* (allele 2), *pen A* (allele 587) and *rpoB* (allele 42). Both *gyrA* 2 and *rpoB* 42 alleles were previously shown to confer no resistance to ciprofloxacin and rifampicin, respectively (*Hong et al., 2013*; *Taha et al., 2007*). Loci NEIS1609 (*folP*) from all isolates, NEIS1600 (*parE*) from NMB_VN2014, and NEIS1753 (*penA*) from NMB_VN2013 and NMB_VN2014 had new allelic variants with no assigned numbers. NEIS1600 and NEIS1753, together with NEIS1635 (*mtrR*), represented three variable loci among the isolates of this study (Table 2). Notably, both allelic variants at *mtrR* locus (7 in NMB_VN2013 and NMB_VN2015, and 1086 in NMB_VN2014) harbored the A39T mutation. This mutation was observed significantly more often in azithromycin-exposed *N. gonorrhoea* carriers (*Wind et al., 2017*), and may result in overexpression of the MtrCDE efflux pump and increased antibiotic resistance in *N. gonorrhoea* (*Demczuk et al., 2017*). Although azithromycin is not recommended by the Vietnam Ministry of Health for the treatment of meningitis, it is recommended by the WHO for the dual therapy (along with ceftriaxone) to treat *N. gonorrhoea* infection. It is thus important to monitor the presence and spreading to potential azithromycin-resistant genetic features in *Neisseria* genus. Other well-known azithromycin-resistant mutations, 23S rRNA A2045G and C2597T, first identified in *N. gonorrhoea* (*Demczuk et al., 2017*), were not found in any Vietnamese isolates in this study which all carried wild-type 23S rRNA.

We confirmed antibiotic susceptibility of NMB_VN2013 and NMB_VN2015 by MIC test, and both strains showed sensitivity to ciprofloxacin, rifampicin, cefotaxime, and ceftriaxone; but diminished susceptibility toward ampicillin, and resistance toward chloramphenicol, though the recorded MICs were much lower than that of NMB_VN2014. While NMB_VN2014 showed reduced susceptibility to rifampicin (MIC = 1.5 μg/ml), both NMB_VN2013 and NMB_VN2015 were still susceptible (MIC = 0.25 and 0.125 μg/ml, respectively) (Table 2).

### Antigenic profiles

Analysis of the deduced peptide sequence at antigenic loci using PubMLST showed all three Vietnamese isolate's genomes carried FHbp Peptide 31, NhbA Peptide 16, and no NadA peptide, but different *porA* and *fetA* variants (Table 3). In detail, NMB_VN2014 and NMB_VN2015 both had *porA* VR1 22-25, but different *porA* VR2, 14-32 and 14, respectively. NMB_VN2013 carried distinct *porA* variants (VR1 7-2, VR2 13) but shared the same *fetA* variant (F4-6) with NMB_VN2014. NMB-VN2015 carried *fetA* variant 1-7. FHbp Peptide 31 belonged to subfamily A, 34 amino acid substituted from Peptide 19 (*U.S. Food and Drug Administration., 2014*) and 97 amino acid differed from Peptide 1 (*U.S. Food and Drug Administration, 2015*). NhbA Peptide 16 contained 79 amino acid substitutions from the Bexsero® component NhbA (Peptide 2). Allele 22-25 of VR1 region of *porA* had eight amino acid substitutions and one deletion compared to allele7-2, and both were not the 1.4 variant used in Bexsero®.

**Table 2** Allelic profiles of antibiotic resistant genes[a] and antibiotic susceptibility of the Vietnamese isolates[b].

| | | | | | | | Locus | | | | | | | Antibiotic susceptibility | | | | | |
|---|---|---|---|---|---|---|---|---|---|---|---|---|---|---|---|---|---|---|---|
| | *gyrA* | *penA* | *rpoB* | NEIS0123 | NEIS0414 | NEIS1320 | NEIS1525 | NEIS1600 | NEIS1609 | NEIS1635 | NEIS1753 | AM | CIP | CTX | CRO | RI | MRP | CL |
| **NMB_VN2012** | 587 | 42 | 1446 | 1 | 32 | 1338 | 1315 | NA* | 7 | NA* | I | S | S | S | S | S | R |
| | | | | | | | | | | | | 0.42 | 0.004 | 0.016 | 0.002 | 0.25 | 0.064 | 62 |
| **NMB_VN2014** | 587 | 42 | 1446 | 1 | 32 | 1338 | NA | NA* | 1086 | NA* | I | S | S | S | I | S | R |
| | | | | | | | | | | | | 0.5 | 0.008 | 0.016 | 0.004 | 1.5 | 0.094 | 256 |
| **NMB_VN2015** | 587 | 42 | 1446 | 1 | 32 | 1338 | 1315 | NA* | 7 | 2242 | I | S | S | S | S | S | R |
| | | | | | | | | | | | | 0.62 | 0.008 | 0.023 | 0.002 | 0.125 | 0.064 | 64 |

**Notes.**

[a] An allele number was assigned to each locus based on its DNA sequence using PubMLST database (https://pubmlst.org/) (*Jolley & Maiden, 2010*).

[b] Antibiotic susceptibility of isolates was examined using E-test strip (bioMerieux, France) and interpreted according to CLSI 2018 breakpoints (*CLSI, 2018*).

NA*, new, unassigned alleles identical at said locus; I, Intermediate; S, Susceptible; R, Resistance.

Numbers below each susceptibility interpretation indicate MIC ($\mu$g/ml) values.

**Table 3** Antigenic profile of Vietnamese and Southeast Asian isolates, with middle line separate the chloramphenicol-resistant (above) and susceptible (below) isolates.

| Isolate | Sequence type and clonal complex (ST (cc))[a] | Antigenic profile[b] | | | | |
|---------|---------|---------|---------|---------|---------|---------|
| | | porA | | fHbp Peptide | nhbA Peptide | fetA |
| | | VR1 | VR2 | | | |
| NMB_VN2013 | 1576 | 7-2 | 13 | 31 | 16 | 4-6 |
| NMB_VN2014 | 13074 | 22-25 | 14-32 | 31 | 16 | 4-6 |
| NMB_VN2015 | 1576 | 22-25 | 14 | 31 | 16 | 1-7 |
| NM01 | 14487 | 19 | 15 | 283 | 16 | 1-20 |
| NM11 | 14496 | 19 | 15 | 31 | – | 3-7 |
| NM12 | 1576 | 19-1 | 15-31 | 31 | 16 | 4-6 |
| NM13 | 1576 | 19 | 15 | – | 16 | 5-88 |
| NM16 | 1576 | 19 | 15 | 31 | – | – |
| NM18 | 1576 | 19 | 15 | 1035 | 16 | 3-31 |
| NM20 | 11005 | 19 | 15-39 | 31 | 16 | 5-135 |
| NM25 | 1576 | 19 | 15 | 31 | 16 | – |
| NM14 | 1145 (cc41/44) | 7-2 | 4 | 14 | 2 | 1-20 |
| NM15 | 41 (cc41/44) | 7-2 | 4 | 14 | 2 | 1-19 |
| NM19 | 14503 (cc4821) | 20 | 23-7 | 141 | 669 | – |
| NM21 | 12811 | 12-1 | 13-1 | 18 | 945 | 1-19 |
| NM23 | 14507 | 22 | 23-1 | – | 21 | 4-21 |
| NM03 | 14488 (cc41/44) | 7-2 | 4 | 14 | 2 | 1-49 |
| NM04 | 14489 | 22-15 | – | 5 | – | – |
| NM06 | 32 (cc32) | 18 | – | 101 | – | 1-21 |
| NM07 | 3256 | 7-1 | – | 24 | 1086 | 3-1 |
| NM09 | 5604 | 22-1 | 26 | – | 1068 | 3-2 |

**Notes.**
[a]Sequence type (ST) and clonal complex (cc) determined by the sequence of seven house-keeping genes (*abcZ, adk, aroE, fumC, gdh, pdhC, and pgm*).
[b]Allele number assigned to each locus based on its DNA (*porA*) or protein (*fHbp, nhbA, and fetA*) sequences.
[c]Analyses was performed using PubMLST database (https://pubmlst.org/) (*Jolley & Maiden, 2010*).

When compared with clinical profiles of the Southeast Asian isolates reported recently by Batty and colleagues (*Batty et al., 2019*), a chloramphenicol-resistant lineage specific features could be observed in sequence types, FHbp Peptide, and NhbA Peptide variants of the Vietnamese and the chloramphenicol-resistant Southeast Asian isolates (Table 3). On the other hand, *fetA* variants showed more variables among groups, and the Vietnamese *porA* loci shared variants with the chloramphenicol-susceptible groups instead of the resistant group.

According to gMATS, a recently developed genotyping tool that predicts strain coverage of 4CMenB (Bexsero®) based on *fHbp, nhbA,* and *porA* VR2 specific genotypes (*Muzzi et al., 2019*), all three Vietnamese isolates were *fHbp* and *porA* non-coverage and *nhbA* unpredictable. Among the other 18 Southeast Asian isolates, three were covered by Bexsero® by all antigenic components (NM03, NM14, and NM15) while one (NM23) was covered by just *nhbA*. Altogether, this resulted in a coverage of 14.3%, 19%, and 14.3% at *fHbp, nhbA,* and *porA* among 21 isolates, respectively. Non-coverage was predicted

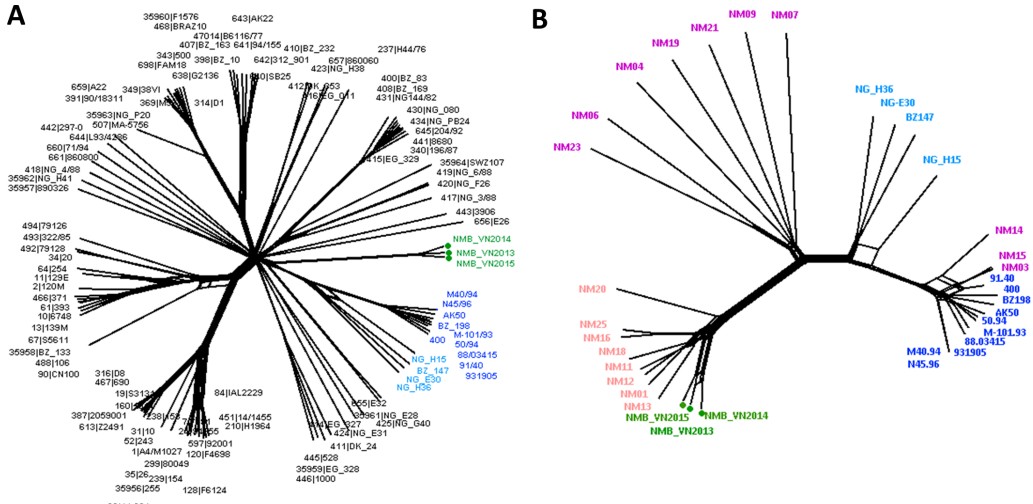

**Figure 1** Genealogical relationship of three Vietnamese isolates with the global 108 *N. meningitidis* isolate collection (*Bratcher et al., 2014*) (A) and the 18 Southeast Asian isolates (*Batty et al., 2019*) (B) revealed by a 1605-locus core genome comparison. These isolates represent the major hyper-invasive lineages/sub-lineages recorded worldwide in the last 70 years and the available genomic information from the Southeast Asia region, respectively. The Vietnamese isolates were highlighted by green nodes and labels in both trees, and major lineages/sub-lineages were indicated by color: navy—ST41 sub-lineage (ten isolates), blue—ST44 sub-lineage (four isolates), pink—Southeast Asian chloramphenicol resistant lineage (eight isolates), purple—Southeast Asian chloramphenicol-susceptible isolates (ten isolates). Together, ST41 and ST44 sub-lineage make up Lineage 3 (ST-41/44 clonal complex).

at 71.4%, 19%, and 85.7% at *fHbp*, *nhbA*, and *porA*, respectively. The rest of variants, including 14.3% of *fHbp*, 61.9% of *nhbA*, and none of *porA* were unpredictable by gMATS. Considering both expression level and genotyping, and extending vaccine coverage prediction to both Trumenba® and Bexsero®, PubMLST antibody cross reactivity also predicted no protection of either vaccine against all three Vietnamese isolates of this study.

## Inference of the Vietnamese isolates' genealogy

Neighbor-Net analysis was used to construct the phylogenetic networks of three Vietnamese isolates and the well-studied 108-isolate global *N. meningitidis* collection (*Bratcher et al., 2014*) based on the seven-locus (MLST), 53-locus (rMLST) and 1605-locus (cgMLST) comparisons. This collection represents the major hyper-invasive lineages/sub-lineages recorded worldwide in the last 70 years. All three methods grouped the Vietnamese isolates into a well-defined, separate clade from the rest of the network. While similar relationships for the Vietnamese isolates were maintained in all analyses, the 1605-locus cgMLST analysis was able to place the Vietnamese branch at the stem of the previously defined Lineage 3 (ST-41/44 clonal complex) (Fig. 1). Additionally, only cgMLST analysis could separate the Vietnamese isolates into three unique strains; while MLST comparison grouped NMB_VN2013 and NMB_VN2015 into one ST, and rMLST identified NMB_VN2013 and NMB_VN2014 as a single strain; showing the close relationship among these strains.

A more refined cluster was observed when the 18 Southeast Asian invasive *N. meningitidis* were added to the genealogical analysis. Eleven chloramphenicol-resistant isolates,
including three Vietnamese isolates, four Thai isolates, and two each from Laos and Cambodia, formed a distinct group, diverged from the rest of Southeast Asian isolates. This lineage seemed to have rapidly expanded clonally in recent years, though this could partially be due to better case report and laboratory detection, since data from this region was scarce up until recently although still remains limited. Although NeighborNet analysis placed this lineage as a divergent branch from other chloramphenicol susceptible isolates in the region, due to the limited number of samples, it remains a possibility for their origin.

The rest of isolates also clustered into two groups, one included NM03, NM14, and NM15 that clustered to the ST-41 sub-lineage of ST-41/44 clonal complex, while the remaining isolates clustered together, seemingly formed a group connecting the chloramphenicol-resistant lineage with ST-44 sub-lineage of ST-41/44 clonal complex (Fig. 1).

## Relationships among Vietnamese isolates revealed by genome comparisons

Of the 1605 loci compared, Genome Comparator identified 1245 identical loci and 355 variable loci in at least one genome of the three isolates in this study. Five loci were missing in all three isolates, four of those were pseudo genes and one encoded for a phage-related protein. Fifteen loci were paralogous loci presumably resulting from assembly of repetitive sequences and thus were excluded from the final analysis. Resultant variable loci were further examined to exclude all variations from genes/pseudo-genes encoding for hypothetical proteins. Consequently, from the initially identified 355 variable loci, we confirmed 264 loci that contained either point mutations, insertions/deletions, or allelic replacements from the three genomes. Among these, a number of sequence variables were observed in adjacent genes, suggesting frequent recombination events between genomes. Pairwise, the genomes of NMB_VN2013 and NMB_VN2014 showed the highest similarity, and NMB_VN2014 and NMB_VN2015 pair showed the lowest. However, pairwise phylogenetic distance between strains calculated from the progressiveMauve alignment using whole genome content was in the range of 0.0158 to 0.0112, indicating a close relationship among them.

Besides core genome comparison, sequences of the known genetic determinants for the virulence of invasive *N. meningitidis*, namely the capsular gene cluster, the Maf-toxin island, and genes encoding for the outer membrane vesicle (OMV) peptides and pilin, from the genomes of three isolates were also compared. The resulting differences, after excluding all variables due to assembly or in the coding sequence of hypothetical proteins, are listed in Table 4. Overall, higher sequence variability could be observed among the three genomes at these loci. From 19 loci of the capsular gene cluster being present for comparison, four confirmed variable loci were *rfbA, cssA, ctrF,* and *cnl.* Except for two assigned alleles found in two loci (allele 113, *cssA* locus of NMB_VN2015's genome and allele 4, *ctrF* locus of NMB_VN2013's genome), the rest of the alleles were newly identified and not yet assigned an allelic number. Four out of 45 loci of the Maf-toxin genomic island included in this comparison, namely *anmK, mafIo2MGI-2, mafA_MGI-3,* and *mafB_MGI-3,* appeared to be different. Sequences of the *anmK* locus of NMB_VN2015 and the *mafB_MGI-3* locus of NMB_VN2014 genome appeared to be novel, unassigned alleles. Detailed nucleotide

**Table 4  Diversity of virulent determining factors from genome sequence of the Vietnamese isolates.**

| Locus | Gene | Product | NMB_VN2013 | NMB_VN2014 | NMB_VN2015 |
|---|---|---|---|---|---|
| *Capsular gene cluster*[a,b] | | | | | |
| NEIS0046 | *rfbA* | Glucose-1-phosphate thymidylyltransferase | NA | NA | NA |
| NEIS0054 | *cssA* | N-acetylglucosamine-6-P 2-epimerase | NA* | NA* | 113 |
| NEIS0067 | *ctrF* | Capsule translocation | 4 | NA* | NA* |
| *Maf-toxin genomic island*[a] | | | | | |
| NEIS1788 | *anmK* | Anhydro-N-acetylmuramic acid kinase | 25 | 25 | NA |
| NEIS1795 | *mafI$_{o2MGI-2}$* | MafI immunity protein | 4;186 | 4 | 4;186 |
| NEIS2083 | *mafA$_{MGI-3}$* | MafA3 lipoprotein | 252 | 252 | 252;890 |
| NEIS2084 | *mafB$_{MGI-3}$* | MafB3 toxin protein | 31 | NA | 31 |
| *Outer membrane vesicle (OMV) peptide*[c] | | | | | |
| NEISp0653 | – | Competence lipoprotein | NA* | NA* | 1 |
| NEISp0275 | – | Organic solvent tolerance protein | NA* | NA | NA* |
| NEISp0923 | – | Antioxidant AhpC TSA family glutaredoxin | 2 | 2 | NA |
| NEISp1364 | – | Outer membrane protein PorA | 155 | NA | NA |
| NEISp1687 | – | Phospholipase A1 | 254 | 254 | NA |
| NEISp1690 | – | Transferrin-binding protein 1 | NA* | NA* | NA |
| NEISp1963 | – | Iron-regulated outer membrane protein FrpB | NA* | NA* | NA |
| *Pilin genes*[a] | | | | | |
| NEIS0020 | *pilB/msrAB* | Peptide methionine sulfoxide reductase MsrA/MsrB | 379 | 11 | NA |
| NEIS0021 | *pilA/ftsY* | Probable signal recognition particle protein | NA | 1778 | 1978 |
| NEIS0036 | *pilT1* | Type IV pilus retraction ATPase PilT | 200 | 11 | 11 |
| NEIS0210 | *pilE* | PilE | NA | NA | NA |
| NEIS0213 | *pglA* | Pilin glycosyltransferase | NA | 1053 | 1053 |
| NEIS0830 | *pilK* | Type IV biogenesis protein | 1905 | 1618 | 1618 |
| pilS | *pilS cassette* | – | NA | NA | NA |

**Notes.**

[a] Allele numbers assigned to each locus based on DNA sequence using PubMLST database (https://pubmlst.org/) (*Jolley & Maiden, 2010*).

[b] Comparison result at *cnl* (capsule null locus) was omitted since all isolates were capsulated.

[c] Allele numbers assigned to each locus based on protein sequence using PubMLST.

NA, new, unassigned alleles; NA*, new, unassigned alleles identical at said locus.

changes analysis revealed a higher percentage of changed nucleotides compared to any other single locus in both *mafA_MGI-3* and *mafB_MGI-3*, which were adjacent to each other, suggesting a recombination event could have happened in this region. Compared to the capsular gene cluster and Maf-toxin island, the CDS of OMV and pilin genes showed higher sequence diversity with variables scattered among loci.

To better understand the probable evolutionary context of these strains, we have further collected and analyzed 25 isolates representing a collection of both historical and later emerged isolates from the same or nearby military units with the three in this study (Table 1). Epidemiological data of these 25 isolates revealed the predominance of serogroup B *N. meningitidis* (78.5%) and the recurrence of two major lineages centered around ST-1576 and ST-4821 in all camp sites (Table 1). The predominance of serogroup B agrees with the previous report from sporadic cases in the region (*Pancharoen et al., 2000*). The other serogroup observed was serogroup C (21.5%). Both ST-1576 and ST-4821

were major hyper-virulent STs with a long history and worldwide distribution. Antigenic profiles associated with these major STs show high similarity to strains described before in China during 1978–2013 (*Zhu et al., 2015*). Besides these, strains representing singleton and novel STs were also frequently observed throughout the years and locations. Available antigenic profiles suggested frequent exchange of genetic material via recombination among strains and reservoirs, with several alleles (P1.20, P1.7-2) associated with the major lineages frequently recurring in different combinations. Clonal expansion was also observed, indicated by the emergence of novel sequence types, and showed no distinct cluster in regard of geological locations, reflecting the close and frequent contact nature of the training units of new military recruits, suggesting close transmission extending beyond individual camps seemed to be the main driving force for *N. meningitidis* prevalence and expansion within the military environment in Vietnam. From searching through the PubMLST database, many emergent STs identified were limited to these units only, highlighting the niche characteristic of *N. meningitidis* population of the military camps.

*N. meningitidis* remains the main cause of sporadic meningitis and sepsis in military recruit camps (*Tran et al., 2019*; *Sim et al., 2013*; *Keiser, Hamilton & Broderick, 2011*). Accurate identification and characterization of the causative strain is crucial for the success of treatment for patients and prophylaxis for contact persons, as well as prevention of outbreaks. Records of Vietnamese and Southeast Asian *N. meningitidis* isolates are still extremely limited, thus it is not possible to determine the origin of these strains, or how they have arisen. Study at the genomic level of additional historical invasive and carriage strains collected at these and nearby camps, or nearby regions can help identify the phylogenetic routes that led to their emergence. On the other hand, since the military setting is among the highest risk group for meningococcal disease in adults, continual effort is needed to provide the surveillance data essential for effective policy making and preparation for response in case of potential outbreaks in the future.

## CONCLUSIONS

In a previous study, we described for the first time the genome of a chloramphenicol-resistant invasive *N. meningitidis* B isolate from a military unit in Vietnam. In this study, using WGS analysis, we characterized the genetic features of two additional *N. meningitidis* B isolates causing sporadic meningitis in another military camp in Vietnam. Core genome comparisons highlight the close phylogenetic relationship of isolates both within and between camps, with emphasis on their shared antibiotic resistant genes and antigenic profiles that are likely not yet covered by current meningococcal B vaccines, Trumenba® and Bexsero®. Another notable shared feature of these isolates was their high resistance against chloramphenicol, likely attributed by but not limited to the 624 bp *catP* variant that were previously found in chloramphenicol-resistant isolates in France (*Galimand et al., 1998*) and Southeast Asia (*Batty et al., 2019*). A phylogenetic network reconstructed from core genome comparison suggests a common lineage of chloramphenicol resistant isolates in the military camps of Vietnam and other Southeast Asian countries that seemed to be expanding in this region.

Since molecular knowledge of the epidemiological characteristics of *N. meningitidis* in Vietnam remains limited, we also reported epidemiological analysis of 25 invasive and carriage strains from these Vietnamese military camps. Besides the major lineages, additional groups of singleton and unique novel sequence types extending beyond individual camps were observed, indicating close transmission is likely the main driving force for *N. meningitidis* prevalence and expansion within the military environment in Vietnam. Taken together, our results provide useful information for further understanding the molecular epidemiology of *N. meningitidis* in the military units in Vietnam, aiding future meningococcal meningitis monitoring and surveillance in the country.

## ACKNOWLEDGEMENTS

We thank members of our laboratories for meaningful discussion and technical assistance. This research utilized the PubMLST database (https://pubmlst.org/) developed by Keith Jolley (*Jolley & Maiden, 2010*).

### Funding

This work is funded by the National Foundation for Science and Technology Development (NAFOSTED) via Grant Numbered 106-NN.02-2015.66 to Huong Minh Nguyen. The funders had no role in study design, data collection and analysis, decision to publish, or preparation of the manuscript.

### Grant Disclosures

The following grant information was disclosed by the authors:
National Foundation for Science and Technology Development (NAFOSTED): 106-NN.02-2015.66.

### Competing Interests

The authors declare there are no competing interests.

### Author Contributions

- Trang Thu Le and Thach Xuan Tran performed the experiments, analyzed the data, prepared figures and/or tables, and approved the final draft.
- Long Phi Trieu performed the experiments, prepared figures and/or tables, authored or reviewed drafts of the paper, and approved the final draft.
- Christopher M. Austin performed the experiments, analyzed the data, authored or reviewed drafts of the paper, and approved the final draft.
- Huong Minh Nguyen conceived and designed the experiments, performed the experiments, analyzed the data, prepared figures and/or tables, authored or reviewed drafts of the paper, and approved the final draft.
- Dong Van Quyen conceived and designed the experiments, authored or reviewed drafts of the paper, and approved the final draft.

## Data Availability

Data is available at NCBI Genomes, BioProject ID PRJNA523495.

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
