# Peer review of "Genotypic characterization and genome comparison reveal insights into potential vaccine coverage and genealogy of Neisseria meningitidis in military camps in Vietnam"

_PeerJ, doi:10.7717/peerj.9502_

## Round 0.1 · original submission · Minor Revisions

The reviewers identified some minor revisions that are required before the manuscript can be considered for publication.

Reviewer 1 ·

Basic reporting

Clear presentation of the results, sufficient introduction provided, relevant references provided, the structure of the article is convenient, tables and figure are clear.

Experimental design

Methods are described with sufficient details and information to replicate.

Validity of the findings

New results of WGS of N. meningitidis in Vietnam are presented, conclusions are well stated.

Additional comments

Abstract, line 29:
I recommend to add the serogoup and the source of isolation for two additional N. meningitidis isolates.

Introduction – line 75:
I recommend using the whole name of the method before using the abbreviation WGS for the first time.

Introduction – lines 83-84:
I recommend to add the serogoup and the source of isolation for two additional N. meningitidis isolates.

Materials and Methods – line 98:
I recommend using Serogroup identification instead of Serogroup typing.

Results – lines 200-202:
It might be useful to add a reference to the source of information about these 108 isolates.

Results – lines 190-191 and 197-198:
I recommend to analyze these results with the gMATS method (Muzzi A. et al., 2019).

Discussion – lines 276-278:
I recommend to discuss the results with the gMATS method (Muzzi A. et al., 2019).

Reviewer 2 ·

Basic reporting

• There are some minor instances throughout where there are slight issues with grammar, which should be corrected before publication.
• There are also issues related to the discussion of genes versus proteins, which are covered in greater detail below. The authors must be clear in their writing in this regard throughout the manuscript.
• There is good background regarding the increased risk for certain communities, which include university students and military units where people live in close proximity. These are well referenced.
• There is good background regarding genomics and MLST for <i>Neisseria meningitidis</i> with appropriate reference to the literature and relevant URLs. This gives the reader sufficient background to understand the context of the work described here.
• The structure of the manuscript needs some attention. For example, the Results includes quite a bit of information that is more appropriate in a Discussion. It might be warranted for the authors to consider creating a combined Results and Discussion, otherwise the more discursive portions of the Results need to be moved to the Discussion, such as in lines 144-145, in lines 159-161, in lines 190-191 (this isn’t clear whether antibody testing in a lab was actually done or this is bioinformatics – if so, it is not in the Methods, in fact even the bioinformatics-based methodology should be in the ), and lines 192-198. This isn’t exhaustive, it is also discursive toward the end of the Results and elsewhere. The manuscript would likely benefit from a combined Results and Discussion.
• In Table 1, it is unclear to the reader what the numbers under each locus represent. It has bene assumed by the reviewer that these are the indicators from MLST, but it is not clear from just looking at the table itself. An appropriate footnote is needed for the reader to be able to understand the nature of this data, citing the PubMLST database.
• In Table 2, the abbreviation ST (cc) should be defined for the reader, so that they are readily able to interpret what this data means in the table and its source. Also, if the antigenic profile is based on the protein, then the protein is properly written as PorA, for example, not <i>PorA</i>. If what has been analysed and indicated by the data in Table 2 is actually the gene information, then it should be <i>porA</i>, starting with a lower case letter, rather than <i>PorA</i> as written here. Again, the source of this data needs to be clarified for the reader with a footnote.
• In Table 3, it would be easier for the general reader in the field to understand what has and has not been found in the isolates if the well-known gene names are included with the Locus information, as well as the Locus (which needs to have a footnote that this is from PubMLST) and Product information. For example, the first three appear to be <i>rfbA</i>, <i>rfbA2</i>, and <i>ctrF</i>. Having deciphered this, it then makes the reader wonder about the rest of the <i>ctr</i> genes in the capsule locus.
• In Table 4, the source of the data that was not generated in this study needs to be indicated in a footnote.
• In Figure 1, the legend should be expanded for the benefit of the reader. Include more detail on how many isolates are in each of the panels and the basis of their inclusion. Also, either remove the red lines or comment upon them in the figure legend. The same is true for the other labels on the trees – remove them or comment upon them and their significance in the legend.
• Tables and Figures need to be able to be able to be understood as stand-alone entities without reference to the main text. Currently, they do not.
• Table 4 is mentioned first in the manuscript (line 106). This should be rectified.
• The data is appropriately deposited as a BioProject with NCBI.

Experimental design

• This research is original, having investigated two isolates from a military unit that is in proximity to one from which <i>N. meningitidis</i> had previously been isolated and sequenced. This provided the authors with some geographic context, whilst expanding out the scope of the previous study. The analysis also included the comparison of their data to the world-wide data available in the PubMLST database and other sources. This is an excellent contribution to the field based on its originality and primary findings.
• The various MLST analyses find that the three Vietnamese isolates cluster into their own separate clade apart from the rest of the network of previously defined <i>N. meningitidis</i>, which is and important result.
• The research question is well defined and clearly presented to the reader.
• There is a good selection of antibiotics that were assessed. It is unfortunate that azithromycin is not one of the antibiotics that was investigated. It is not something this reviewer would ask that the investigators go back to the laboratory and check – particularly in the current environment when so many labs are closed, but even if this were not the case. This is a choice that was made by the investigators and that is fine. Given the WHO recommendations for dual therapy with ceftriaxone and azithromycin, however, it might have been interesting to know. Perhaps the authors could look at the potential genomic features that may contribute to resistance or otherwise comment in the manuscript on azithromycin in the absence of this testing.
• It must be clear in the use of language when the investigation has based its conclusions on genomic data and on experimental data. For example, in lines 180-181, the authors discuss expression of antigenic proteins. This cannot be known or determined from analysis of the genomic data. Unless there were experimental investigations undertaken using immunoassays that are not included in the Methods, this sentence and the presented information needs to be considerably re-worded and presented accurately to the readers. Possessing genetic sequences does not necessarily equate to expression of a protein. The authors must be clear when writing that use of the protein name, such a PorA, implies that they have investigated the protein. From the Methods, they have not. They have done a genomic analysis and therefore can comment only upon the sequence information about <i>porA</i>. Great care and attention needs to be paid to how bioinformatics investigations are written about and presented to the reader.
• In the capsular cluster investigation, is cnl not the designation for the capsule null locus – as in the genomic location when there are no capsular genes, present in some isolates of <i>N. meningitidis</i>? This is not clear from the manuscript as written and should be made so for the reader. Indeed, mention of <i>cnl</i> could possibly actually be removed because the two isolates have capsule loci, therefore there is no need to even compare to the <i>cnl</i>. An appropriate comparison is therefore to capsular isolates.
• Methods are described with sufficient detail either in this manuscript or in the references cited. The methods being used are appropriate for this investigation.

Validity of the findings

• The novelty of the data has been well assessed, although there are some points that need to be clarified and written in a more robust manner, as pointed out in the other sections of this review.
• The genomic data assessment on the general features are robust and correct that they align with the previous genomes and are typical. However, the authors should include a reference to previous literature for the benefit of readers. Reference to some of the early complete genome sequences for <i>N. meningitidis</i> and <i>N. gonorrhoeae</i> from mid-2000s would be sufficient.
• It isn’t clear on what basis capsule switching is proposed (lines 259-260). Also, capsule switching as a process should be cited.
• The conclusions are clearly and well stated and link directly to the findings of the research. They are supported by the data, especially because they include that the data predicts that the isolates are not covered by the serogroup B vaccines, since this is based on bioinformatics analysis.

Additional comments

• The title is accurate, however it does not address the most striking finding of this paper, which is that the isolates that were sequenced from the Vietnamese military units were predicted to not be covered by either the Bexsero or Trumenba vaccines. This is a significant discovery that might be overlooked by researchers skimming the literature by title. The authors are urged to consider a title that reflects this finding and highlights it, therefore directing readers to their publication.
• The term <i>de novo</i>, as first used in the Abstract (line 33), should be in italics as a non-English term.
• The gene names should also be in italics, such as <i>aroE</i> (twice on line 155).
• In line 222, ‘pseudo’ is not a word in itself. It should be hyphenated as pseudo-gene or written as pseudogene.
• In line 268, <i>Neisseria</i> should be abbreviated. This is not the first instance of its use in the manuscript.
• Line 278 has an undefined abbreviation ‘NmB’ that should be defined and clarified for the benefit of all readers.
• Genus species names and gene names in the references should be italics.
• The gene names in Table 1, <i>gyrA</i>, <i>penA</i>, and <i>rpoB</i> should be in italics.

---

## Round 0.2 · Minor Revisions

I have added a couple of gramatical edits using track changes (mine are in red), including the section between 199 and 211. I think you need to write these as protein names as you are talking about the peptide sequences, i realise this was changed in response to a reviewer comment and that they are deduced for the genomic sequence, but you have made this clear at the beginning of the paragraph

---

## Round 0.3 · accepted · Accept

Thanks for your manuscript.